# RFOLD: RNA SECONDARY STRUCTURE PREDICTION WITH DECOUPLED OPTIMIZATION

## ABSTRACT

The secondary structure of ribonucleic acid (RNA) is more stable and accessible in the cell than its tertiary structure, making it essential for functional prediction. Although deep learning has shown promising results in this field, current methods suffer from poor generalization and high complexity. In this work, we present RFold, a simple yet effective RNA secondary structure prediction in an end-to-end manner. RFold introduces a decoupled optimization process that decomposes the vanilla constraint satisfaction problem into row-wise and column-wise optimization, simplifying the solving process while guaranteeing the validity of the output. Moreover, RFold adopts attention maps as informative representations instead of designing hand-crafted features. Extensive experiments demonstrate that RFold achieves competitive performance and about eight times faster inference efficiency than the state-of-the-art method.

## 1 INTRODUCTION

Ribonucleic acid is essential in structural biology for its diverse functional classes (Geisler & Coller, 2013). The functions of RNA molecules are determined by their structure (Sloma & Mathews, 2016). The secondary structure, which contains the nucleotide base pairing information, as shown in Fig. 1, is crucial for the correct functions of RNA molecules (Fallmann et al., 2017). Although experimental assays such as X-ray crystallography (Cheong et al., 2004), nuclear magnetic resonance (Fürtig et al., 2003), and cryogenic electron microscopy (Fica & Nagai, 2017) can be implemented to determine RNA secondary structure, they suffer from low throughput and expensive cost.

Computational RNA secondary structure prediction methods have become increasingly popular due to their high efficiency (Iorns et al., 2007). Currently, these methods can be broadly classified into two categories (Rivas, 2013; Fu et al., 2022; Singh et al., 2019; Szikszai et al., 2022): (i) comparative sequence analysis and (ii) single sequence folding algorithm. Comparative sequence analysis determines the secondary structure conserved among homologous sequences but the limited known RNA families hinder its development (Knudsen & Hein, 2003; Hofacker et al., 2002; Gutell et al., 2002; Griffiths-Jones et al., 2003; Gardner et al., 2009; Nawrocki et al., 2015). Researchers thus resort to single RNA sequence folding algorithms that do not need multiple sequence alignment information. A classical category of computational RNA folding

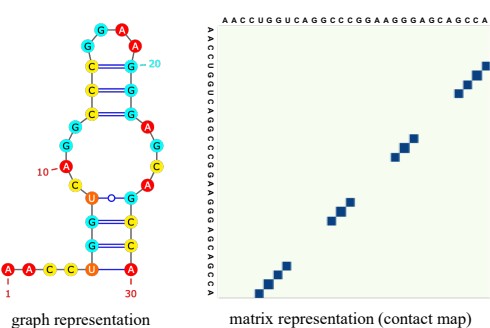

graph representation          matrix representation (contact map)

Figure 1: The graph and matrix representation of an RNA secondary structure example.

algorithms is to use dynamic programming (DP) that assumes the secondary structure is a result of energy minimization (Bellaousov et al., 2013; Lorenz et al., 2011; Do et al., 2006). However, energy-based approaches usually require a nested structure, which ignores some biologically essential structures such as pseudoknots, i.e., non-nested base pairs (Chen et al., 2019; Seetin & Mathews, 2012; Xu & Chen, 2015), as shown in Fig. 2. Since predicting secondary structures with pseudoknots under the energy minimization framework has shown to be hard and NP-complete (Wang & Tian, 2011; Fu et al., 2022), deep learning techniques are introduced as an alternative approach.

Attempts to overcome the limitations of energy-based methods have motivated deep learning methods in the absence of DP. The general deep-learning-based RNA secondary structure prediction methods can be decomposed into three key parts:

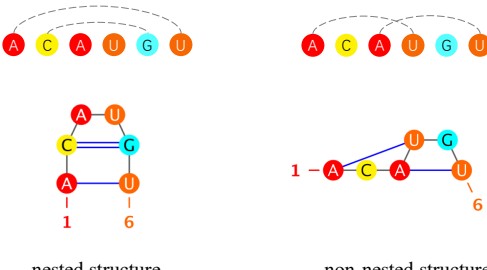

Figure 2: Examples of nested and non-nested secondary structures.

- *pre-processing*: projecting 1D sequence into 2D matrix. (**1D → discrete/continuous 2D**)

- *backbone model*: learning from the 2D matrix and then outputs a hidden matrix of continuous values. (**discrete/continuous 2D → continuous 2D**)

- *post-processing*: converting the hidden matrix into a contact map, which is a matrix of discrete 0/1 values. (**continuous 2D → discrete 2D**)

SPOT-RNA (Singh et al., 2019) is a seminal work that ensembles ResNet and LSTM to identify molecular features. SPOT-RNA does not constrain the output space into valid structures, which degrades its generalization (Jung et al.). E2Efold (Chen et al., 2019) employs an unrolled algorithm that post-processes the network output to satisfy the constraints. E2Efold introduces a convex relaxation to make the optimization tractable, leading to possible constraint violations and poor generalization ability (Sato et al., 2021). Developing an appropriate optimization that forces the output to be valid becomes an important issue. Apart from the optimization problem, state-of-the-art approaches require hand-crafted features and introduce the pre-processing step for such features, which is inefficient and needs expert knowledge. CDPfold (Zhang et al., 2019) develops a matrix representation based on sequence pairing that reflects the implicit matching between bases. UFold (Fu et al., 2022) follows the exact post-process mechanism as E2Efold and uses hand-crafted features from CDPfold with U-Net (Ronneberger et al., 2015) model architecture to improve the performance.

Although promising, current deep learning methods on RNA secondary structure prediction have been distressed by: (1) *the post-processing optimization that is complicated and poor in generalization* and (2) *the pre-processing that requires expensive complexity and expert knowledge*. In this work, we present RFold, a simple yet effective RNA secondary structure prediction method. Specifically, we introduce a decoupled optimization process that decomposes the vanilla constraint satisfaction problem into row-wise and column-wise optimization, simplifying the solving process while guaranteeing the validity of the output. Besides, we adopt attention maps as informative representations to automatically learn the pair-wise interactions of the nucleotide bases instead of using hand-crafted features to perform data pre-processing. The methodological comparison between RFold and other classical methods is summarized in Appendix A. Extensive experiments to compare RFold with state-of-the-art methods on several benchmark datasets and show the superior performance of our proposed method. Moreover, RFold has faster inference efficiency due to its simplicity.

## 2 RELATED WORK

**Comparative Sequence Analysis** Comparative sequence analysis determines base pairs conserved among homologous sequences (Gardner & Giegerich, 2004; Knudsen & Hein, 2003; Gorodkin et al., 2001). ILM (Ruan et al., 2004) combines thermodynamic and mutual information content scores. Sankoff (Hofacker et al., 2004) merges the sequence alignment and maximal-pairing folding methods (Nussinov et al., 1978). Dynalign (Mathews & Turner, 2002) and Carnac (Touzet & Perriquet, 2004) are the subsequent variants of Sankoff algorithms. RNA forester (Hochsmann et al., 2003) introduces a tree alignment model for global and local alignments. However, the limited number of known RNA families (Nawrocki et al., 2015) impedes the development of comparative methods.

**Energy-based Folding Algorithms** When the secondary structure consists only of nested base pairing, dynamic programming can efficiently predict the structure by minimizing energy. Early works include Vienna RNAfold (Lorenz et al., 2011), Mfold (Zuker, 2003), RNAstructure (Mathews & Turner, 2006), and CONTRAfold (Do et al., 2006). Faster implementations that speed up dynamic programming have been proposed, such as Vienna RNAplfold (Bernhart et al., 2006), LocalFold (Lange et al., 2012), and LinearFold (Huang et al., 2019). However, these methods cannot accurately predict structures with pseudoknots, as predicting the lowest free energy structures with pseudoknots is NP-complete (Lyngsø & Pedersen, 2000), making it difficult to improve performance.

**Learning-based Folding Algorithms** SPOT-RNA (Singh et al., 2019) is a seminal work that employs deep learning for RNA secondary structure prediction. SPOT-RNA2 (Singh et al., 2021) improves its predecessor by using evolution-derived sequence profiles and mutational coupling. Inspired by Raptor-X (Wang et al., 2017) and SPOT-Contact (Hanson et al., 2018), SPOT-RNA uses ResNet and bidirectional LSTM with a sigmoid function to output the secondary structures. MXfold (Akiyama et al., 2018) is also an early work that combines support vector machines and thermodynamic models. CDPfold (Zhang et al., 2019), DMFold (Wang et al., 2019), and MXFold2 (Sato et al., 2021) integrate deep learning techniques with energy-based methods. E2Efold (Chen et al., 2019) takes a remarkable step in constraining the output to be valid by learning unrolled algorithms. However, its relaxation for making the optimization tractable may violate the structural constraints. UFold (Fu et al., 2022) further introduces U-Net model architecture to improve performance.

## 3 PRELIMINARIES AND BACKGROUNDS

### 3.1 PRELIMINARIES

The primary structure of RNA is the ordered linear sequence of bases, which is typically represented as a string of letters. Formally, an RNA sequence can be represented as $\boldsymbol{X} = (x_1, ..., x_L)$, where $x_i \in \{A, U, C, G\}$ denotes one of the four bases, i.e., *Adenine* (A), *Uracil* (U), *Cytosine* (C), and *Guanine* (G). The secondary structure of RNA is a contact map represented as a matrix $\boldsymbol{M} \in \{0, 1\}^{L \times L}$, where $\boldsymbol{M}_{ij} = 1$ if the $i$-th and $j$-th bases are paired. In the RNA secondary structure prediction problem, we aim to obtain a model with learnable parameters $\Theta$ that learns a mapping $\mathcal{F}_{\Theta} : \boldsymbol{X} \mapsto \boldsymbol{M}$ by exploring the interactions between bases. Here, we decompose the mapping $\mathcal{F}_{\Theta}$ into two sub-mappings as:

$$\mathcal{F}_{\Theta} := \mathcal{G}_{\theta_g} \circ \mathcal{H}_{\theta_h}, \tag{1}$$

where $\mathcal{H}_{\theta_h} : \boldsymbol{X} \mapsto \boldsymbol{H}$, $\mathcal{G}_{\theta_g} : \boldsymbol{H} \mapsto \boldsymbol{M}$ are mappings parameterized by $\theta_h$ and $\theta_g$, respectively. $\boldsymbol{H} \in \mathbb{R}^{L \times L}$ is regarded as the unconstrained output of neural networks.

### 3.2 BACKGROUND

It is worth noting that there are hard constraints on the formation of RNA secondary structure, meaning that certain types of pairing are not available (Steeg, 1993). Such constraints (Chen et al., 2019) can be formally described as follows:

- (a) Only three types of nucleotide combinations can form base pairs: $\mathcal{B} := \{AU, UA\} \cup \{GC, CG\} \cup \{GU, UG\}$. For any base pair $x_i x_j$ where $x_i x_j \notin \mathcal{B}$, $\boldsymbol{M}_{ij} = 0$.
- (b) No sharp loops within three bases. For any adjacent bases, there can be no pairing between them, i.e., $\forall |i - j| \leqslant 3, \boldsymbol{M}_{ij} = 0$.
- (c) There can be at most one pair for each base, i.e., $\forall i, \sum_{j=1}^{L} \boldsymbol{M}_{ij} \leqslant 1$.

The available space of valid secondary structures is all *symmetric* matrices $\in \{0, 1\}^{L \times L}$ that satisfy the above three constraints. The first two constraints can be satisfied easily. We define a constraint matrix $\overline{\boldsymbol{M}}$ as: $\overline{\boldsymbol{M}}_{ij} := 1$ if $x_i x_j \in \mathcal{B}$ and $|i - j| \geqslant 4$, and $\overline{\boldsymbol{M}}_{ij} := 0$ otherwise. By element-wise multiplication of the network output and the constraint matrix $\overline{\boldsymbol{M}}$, invalid pairs are masked.

The critical issue in obtaining a valid RNA secondary structure is the third constraint, i.e., *processing the network output to create a symmetric binary matrix that only allows a single "1" to exist in each row and column*. There are different strategies for dealing with this issue.

**SPOT-RNA** is a typical kind of method that imposes minor constraints. It takes the original output of neural networks $\boldsymbol{H}$ and directly applies the Sigmoid function, assigning a value of 1 to those greater than 0.5 and 0 to those less than 0.5. This process can be represented as:

$$\mathcal{G}(\boldsymbol{H}) = \mathbb{1}_{[\text{Sigmoid}(\boldsymbol{H}) > 0.5]} \odot \boldsymbol{H}. \tag{2}$$

Here, the offset term $s$ has been set to 0.5. No explicit constraints are imposed, and no additional parameters $\theta_g$ are required.

**E2Efold** formulates the problem with constrained optimization and introduces an intermediate variable $\widehat{M} \in \mathbb{R}^{L \times L}$. It aims to maximize the predefined score function:

$$\mathcal{S}(\widehat{M}, H) = \frac{1}{2}\left\langle H - s, \mathcal{T}(\widehat{M})\right\rangle - \rho\|\widehat{M}\|_1, \tag{3}$$

where $\mathcal{T}(\widehat{M}) = \frac{1}{2}(\widehat{M} \odot \widehat{M} + (\widehat{M} \odot \widehat{M})^T) \odot \overline{M}$ ensures the output is a symmetric matrix that satisfies the constraints (a-b), $s$ is an offset term that is set as $\log(9.0)$ here, $\langle \cdot, \cdot \rangle$ denotes matrix inner product and $\rho\|\widehat{M}\|_1$ is a $\ell_1$ penalty term to make the matrix to be sparse.

The constraint (c) is imposed by requiring Eq. 3 to satisfy $\mathcal{T}(\widehat{M})\mathbb{1} \leqslant \mathbb{1}$. Thus, Eq. 3 is rewritten as:

$$\mathcal{S}(\widehat{M}, H) = \min_{\boldsymbol{\lambda} \geqslant \mathbf{0}} \frac{1}{2}\left\langle H - s, \mathcal{T}(\widehat{M})\right\rangle - \rho\|\widehat{M}\|_1 \quad -\left\langle \boldsymbol{\lambda}, \mathrm{ReLU}(\mathcal{T}(\widehat{M})\mathbb{1} - \mathbb{1})\right\rangle, \tag{4}$$

where $\boldsymbol{\lambda} \in \mathbb{R}_+^L$ is a Lagrange multiplier.

Formally, this process can be represented as:

$$\mathcal{G}_{\theta_g}(H) = \mathcal{T}(\arg\max_{\widehat{M} \in \mathbb{R}^{L \times L}} \mathcal{S}(\widehat{M}, H)). \tag{5}$$

Though three constraints are explicitly imposed in E2Efold, this method requires iterative steps to approximate the valid solutions and cannot guarantee that the results are entirely valid. Moreover, it needs a set of parameters $\theta_g$ in this processing, making tuning the model complex.

## 4 RFOLD

### 4.1 DECOUPLED OPTIMIZATION

We propose the following formulation for the constrained optimization problem in RNA secondary structure problem:

$$\min_{M} - \mathrm{tr}(M^T\widehat{H})$$

$$\text{s.t.} \sum_{j=1}^{L} M_{ij} \leqslant 1, \forall i; \sum_{i=1}^{L} M_{ij} \leqslant 1, \forall j, \tag{6}$$

where $\mathrm{tr}(M^T\widehat{H}) = \sum_{i=1}^{L}\sum_{j=1}^{L} M_{ij}\widehat{H}_{ij}$ represents the trace operation. The matrix $\widehat{H}$ is symmetrized based on the original network output $H$ while satisfying the constraints (a-b) in Sec. 3.2 by multiplying the constraint matrix $\overline{M}$, i.e., $\widehat{H} = (H \odot H^T) \odot \overline{M}$.

We then propose to decouple the optimization process into row-wise and column-wise optimizations, and define the corresponding selection schemes as $S_r$ and $S_c$ respectively:

$$S_r = \{S_r^1, S_r^2, ..., S_r^L\}, \ S_c = \{S_c^1, S_c^2, ..., S_c^L\}, \tag{7}$$

where $S_r^i \in \{0, 1\}^L$ signifies the selection scheme on the $i$th row, and $S_c^j \in \{0, 1\}^L$ represents the selection scheme on the $j$th column. The score function is defined as:

$$\mathcal{S}(S_r, S_c, \widehat{H}) = -\mathrm{tr}(M^T\widehat{H}), \tag{8}$$

where $S_r, S_c$ constitute the decomposition of $M$. The goal of the score function is to maximize the dot product of $M$ and $\widehat{H}$ in order to select the maximum value in $\widehat{H}$. Our proposed decoupled optimization reformulates the original constrained optimization problem in Equation 6 as follows:

$$\min_{S_r, S_c} \mathcal{S}(S_r, S_c)$$

$$\text{s.t.} \sum_{i=1}^{L} S_r^i \leqslant \mathbb{1}, \forall r; \sum_{j=1}^{L} S_c^j \leqslant \mathbb{1}, \forall c. \tag{9}$$

If the corresponding $\widehat{H}_{ij}$ have the highest score in its row $\{\widehat{H}_{ik}\}_{k=1}^L$ and its column $\{\widehat{H}_{kj}\}_{k=1}^L$, then $M_{ij} = 1$. By exploring the optimal $S_r$ and $S_c$, the chosen base pairs can be obtained by the a greedy algorithm solution $S = S_r \otimes S_c$.

### 4.2 ROW-COL ARGMAX

With the proposed decoupled optimization, the optimal matrix can be easily obtained using the variant Argmax function:

$$\text{Row-Col-Argmax}(\widehat{\boldsymbol{H}}) = \text{Row-Argmax}(\widehat{\boldsymbol{H}}) \odot \text{Col-Argmax}(\widehat{\boldsymbol{H}}) \tag{10}$$

where Row-Argmax and Col-Argmax are row-wise and column-wise Argmax functions respectively:

$$\text{Row-Argmax}_{ij}(\widehat{\boldsymbol{H}}) = \begin{cases} 1, & \text{if } \max\{\widehat{\boldsymbol{H}}_{ik}\}_{k=1}^{L} = \widehat{\boldsymbol{H}}_{ij}, \\ 0, & \text{otherwise.} \end{cases}$$

$$\text{Col-Argmax}_{ij}(\widehat{\boldsymbol{H}}) = \begin{cases} 1, & \text{if } \max\{\widehat{\boldsymbol{H}}_{kj}\}_{k=1}^{L} = \widehat{\boldsymbol{H}}_{ij}, \\ 0, & \text{otherwise.} \end{cases} \tag{11}$$

**Theorem 1.** Given a symmetric matrix $\widehat{\boldsymbol{H}} \in \mathbb{R}^{L \times L}$, the matrix $\text{Row-Col-Argmax}(\widehat{\boldsymbol{H}})$ is also a symmetric matrix.

*Proof:* See Appendix E.1.

As shown in Fig. 3, taking a random symmetric $6 \times 6$ matrix as an example, we show the output matrics of Row-Argmax, Col-Argmax, and Row-Col-Argmax functions, respectively. The Row-Col Argmax selects the value that has the maximum value on both its row and column while keeping the output matrix symmetric.

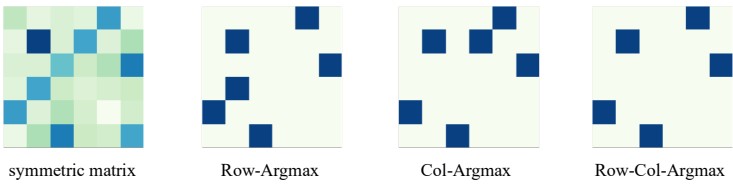

symmetric matrix     Row-Argmax     Col-Argmax     Row-Col-Argmax

Figure 3: The visualization of the Row-Col-Argmax function.

From Theorem 1, we can observe that $\text{Row-Col-Argmax}(\widehat{\boldsymbol{H}})$ is a symmetric matrix that satisfies the constraint (c). Since $\widehat{\boldsymbol{H}}$ already satisfies constraints (a-b), the optimized output is:

$$\mathcal{G}(\boldsymbol{H}) = S_r \otimes S_c = \text{Row-Col-Argmax}(\widehat{\boldsymbol{H}}), \tag{12}$$

where $S_r, S_c = \arg\min_{S_r, S_c} -\text{tr}(S_r, S_c)$.

### 4.3 ROW-COL SOFTMAX

Though the Row-Col Argmax function can obtain the optimal matrix $\mathcal{G}(\boldsymbol{H})$, it is not differentiable and thus cannot be directly used in the training process. In the training phase, we need to use a differentiable function to approximate the optimal results. Therefore, we propose using a Row-Col Softmax function to approximate the Row-Col Argmax function for training. To achieve this, we perform row-wise Softmax and column-wise Softmax on the symmetric matrix $\widehat{\boldsymbol{H}}$ separately, as shown below:

$$\text{Row-Softmax}_{ij}(\widehat{\boldsymbol{H}}) = \frac{\exp(\widehat{\boldsymbol{H}}_{ij})}{\sum_{k=1}^{L} \exp(\widehat{\boldsymbol{H}}_{ik})},$$

$$\text{Col-Softmax}_{ij}(\widehat{\boldsymbol{H}}) = \frac{\exp(\widehat{\boldsymbol{H}}_{ij})}{\sum_{k=1}^{L} \exp(\widehat{\boldsymbol{H}}_{kj})}. \tag{13}$$

The Row-Col Softmax function is then defined as follows:

$$\text{Row-Col-Softmax}(\widehat{\boldsymbol{H}}) = \frac{1}{2}(\text{Row-Softmax}(\widehat{\boldsymbol{H}}) + \text{Col-Softmax}(\widehat{\boldsymbol{H}})), \tag{14}$$

Note that we use the average of $\text{Row-Softmax}(\widehat{\boldsymbol{H}})$ and $\text{Col-Softmax}(\widehat{\boldsymbol{H}})$ instead of the element product as shown in Eq. 10 for the convenience of optimization.

**Theorem 2.** Given a symmetric matrix $\widehat{\boldsymbol{H}} \in \mathbb{R}^{L \times L}$, the matrix Row-Col-Softmax$(\widehat{\boldsymbol{H}})$ is also a symmetric matrix.

*Proof:* See Appendix E.2.

As shown in Fig. 4, taking a random symmetric $6 \times 6$ matrix as an example, we show the output matrics of Row-Softmax, Col-Softmax, and Row-Col-Softmax functions, respectively. It can be seen that the output matrix of Row-Col-Softmax is still symmetric. Leveraging the differentiable property of Row-Col-Softmax, the model can be easily optimized.

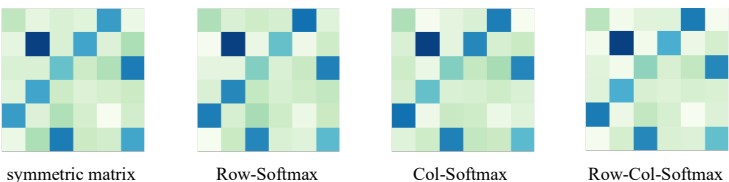

symmetric matrix      Row-Softmax      Col-Softmax      Row-Col-Softmax

Figure 4: The visualization of the Row-Col-Softmax function.

In the training phase, we apply the differentiable Row-Col Softmax activation and optimize the mean square error (MSE) loss function between $\mathcal{G}(\boldsymbol{H})$ and $\boldsymbol{M}$:

$$\mathcal{L}(\mathcal{G}(\boldsymbol{H}), \boldsymbol{M}) = \frac{1}{L^2} \|\text{Row-Col-Softmax}(\widehat{\boldsymbol{H}}) - \boldsymbol{M}\|^2. \tag{15}$$

### 4.4 SEQ2MAP ATTENTION

To simplify the pre-processing step that constructs hand-crafted features based on RNA sequences, we propose a Seq2map attention module that can automatically produce informative representations. We start with a sequence in the one-hot form $\boldsymbol{X} \in \mathbb{R}^{L \times 4}$ and obtain the sum of the token embedding and positional embedding as the input for the Seq2map attention. For convenience, we denote the input as $\boldsymbol{Z} \in \mathbb{R}^{L \times D}$, where $D$ is the hidden layer size of the token and positional embeddings.

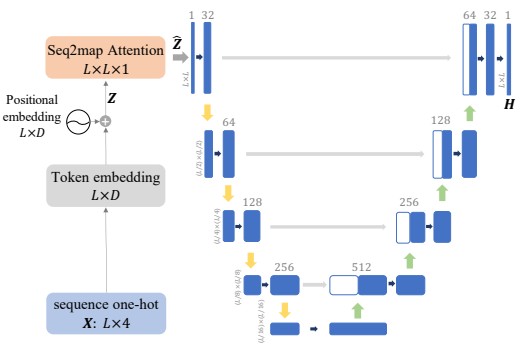

Motivated by the recent progress in attention mechanisms (Vaswani et al., 2017; Choromanski et al., 2020; Katharopoulos et al., 2020; Hua et al., 2022), we aim to develop a highly effective sequence-to-map transformation based on pair-wise attention. We obtain the query

Figure 5: The overview model of RFold.

$\boldsymbol{Q} \in \mathbb{R}^{L \times D}$ and key $\boldsymbol{K} \in \mathbb{R}^{L \times D}$ by applying per-dim scalars and offsets to $\boldsymbol{Z}$:

$$\boldsymbol{Q} = \gamma_Q \boldsymbol{Z} + \beta_Q, \ \boldsymbol{K} = \gamma_K \boldsymbol{Z} + \beta_K, \tag{16}$$

where $\gamma_Q, \gamma_K, \beta_Q, \beta_K \in \mathbb{R}^{L \times D}$ are learnable parameters.

Then, the pair-wise attention map is obtained by:

$$\bar{\boldsymbol{Z}} = \text{ReLU}^2(\boldsymbol{Q}\boldsymbol{K}^T/L), \tag{17}$$

where $\text{ReLU}^2$ is an activation function that can be recognized as a simplified Softmax function in vanilla Transformers (So et al., 2021). The output of Seq2map is the gated representation of $\bar{\boldsymbol{Z}}$:

$$\widehat{\boldsymbol{Z}} = \bar{\boldsymbol{Z}} \odot \sigma(\bar{\boldsymbol{Z}}), \tag{18}$$

where $\sigma(\cdot)$ is the Sigmoid function that performs as a gate operation.

As shown in Fig. 5, we identify the problem of predicting $\boldsymbol{H} \in \mathbb{R}^{L \times L}$ from the given sequence attention map $\widehat{\boldsymbol{Z}} \in \mathbb{R}^{L \times L}$ as an image-to-image segmentation problem and apply the U-Net model architecture to extract pair-wise information.

## 5 EXPERIMENTS

We conduct experiments to compare our proposed RFold with state-of-the-art and commonly used approaches. Multiple experimental settings are taken into account, including standard secondary structure prediction, generalization evaluation, large-scale benchmark evaluation, and inference time comparison. Detailed descriptions of the experimental setups can be found in the Appendix B.

### 5.1 STANDARD RNA SECONDARY STRUCTURE PREDICTION

Following (Chen et al., 2019), we split the RNAStralign dataset into train, validation, and test sets by stratified sampling. We report the results in Table 1. Energy-based methods achieve relatively weak F1 scores ranging from 0.420 to 0.633. Learning-based folding algorithms like E2Efold and UFold significantly improve performance by large margins, while RFold obtains even better performance among all the metrics. Moreover, RFold obtains about 8% higher precision than the state-of-the-art method. This suggests that our decoupled optimization is strict to satisfy all the hard constraints for predicting valid structures. Results on RNA structures with pseudoknots are shown in Appendix C.

Table 1: Results on RNAStralign test set. Results in bold and underlined are the top-1 and top-2 performances, respectively.

| Method | Precision | Recall | F1 |
|---|---|---|---|
| Mfold | 0.450 | 0.398 | 0.420 |
| RNAfold | 0.516 | 0.568 | 0.540 |
| RNAstructure | 0.537 | 0.568 | 0.550 |
| CONTRAfold | 0.608 | 0.663 | 0.633 |
| LinearFold | 0.620 | 0.606 | 0.609 |
| CDPfold | 0.633 | 0.597 | 0.614 |
| E2Efold | 0.866 | 0.788 | 0.821 |
| UFold | 0.905 | 0.927 | 0.915 |
| RFold | **0.981** | **0.973** | **0.977** |

### 5.2 GENERALIZATION EVALUATION

To verify the generalization ability, we evaluate the performance on another benchmark dataset ArchiveII using the pre-trained model on the RNAStralign training dataset. Following (Chen et al., 2019), we exclude RNA sequences in ArchiveII that have overlapping RNA types with the RNAStralign dataset for a fair comparison. The results are reported in Table 2. Among the state-of-the-art methods, RFold attains the highest F1 score. It is noteworthy that RFold has a relatively lower recall metric and significantly higher precision metric. This phenomenon may be due to the strict constraints imposed by RFold. It may cover fewer pairwise interactions, resulting in a lower recall. Nonetheless, the highest F1 score indicates the excellent generalization ability of RFold.

Moreover, we assess performance using the bpRNA-new dataset to further validate generalizability on cross-family data, with results depicted in Table 3. Pure deep learning methods have struggled with this task. UFold, for instance, relies on the thermodynamic method for data augmentation to achieve satisfactory results. Notably, UFold achieves an F1 score of 0.583, while RFold reaches 0.616. When the data augmentation strategy based on Contrafold is incorporated, the performance of UFold escalates to 0.636, while our RFold method realizes a score of 0.651. This positions RFold as second only to the thermodynamics-based method, Contrafold, in terms of F1 score.

Table 2: Results on ArchiveII dataset.

| Method | Precision | Recall | F1 |
|---|---|---|---|
| Mfold | 0.668 | 0.590 | 0.621 |
| RNAfold | 0.663 | 0.613 | 0.631 |
| RNAstructure | 0.664 | 0.606 | 0.628 |
| CONTRAfold | 0.696 | 0.651 | 0.665 |
| LinearFold | 0.724 | 0.605 | 0.647 |
| RNAsoft | 0.665 | 0.594 | 0.622 |
| Eternafold | 0.667 | 0.622 | 0.636 |
| E2Efold | 0.734 | 0.660 | 0.686 |
| SPOT-RNA | 0.743 | 0.726 | 0.711 |
| MXfold2 | 0.788 | 0.760 | 0.768 |
| Contextfold | 0.873 | 0.821 | 0.842 |
| UFold | 0.887 | **0.928** | 0.905 |
| RFold | **0.938** | 0.910 | **0.921** |

Table 3: Results on bpRNA-new dataset.

| Method | Precision | Recall | F1 |
|---|---|---|---|
| Mfold | 0.584 | 0.692 | 0.623 |
| RNAfold | 0.593 | 0.720 | 0.640 |
| RNAstructure | 0.586 | 0.704 | 0.629 |
| CONTRAfold | **0.620** | 0.736 | **0.661** |
| LinearFold | 0.658 | 0.645 | 0.633 |
| RNAsoft | 0.580 | 0.692 | 0.620 |
| Externafold | 0.598 | 0.732 | 0.647 |
| SPOT-RNA | 0.635 | 0.641 | 0.620 |
| MXfold2 | 0.599 | 0.715 | 0.641 |
| Contextfold | 0.596 | 0.636 | 0.604 |
| UFold | 0.500 | 0.736 | 0.583 |
| UFold + aug | 0.570 | **0.742** | 0.636 |
| RFold | 0.614 | 0.619 | 0.616 |
| RFold + aug | 0.618 | 0.687 | 0.651 |

## 5.3 LARGE-SCALE BENCHMARK EVALUATION

The large-scale benchmark bpRNA has a fixed training set (TR0), evaluation set (VL0), and testing set (TS0). Following (Singh et al., 2019; Fu et al., 2022), we train the model in bpRNA-TR0 and evaluate the performance on bpRNA-TS0 by using the best model learned from bpRNA-VL0. We summarize the evaluation results on bpRNA-TS0 in Table 4. It can be seen that RFold significantly improves the previous state-of-the-art method SPOT-RNA by 4.0% in the F1 score.

Following (Fu et al., 2022), we conduct an experiment on long-range interactions. The bpRNA-TS0 dataset contains more versatile RNA sequences of different lengths and various types, which can be a reliable evaluation. Given a sequence of length $L$, the long-range base pairing is defined as the paired and unpaired bases with intervals longer than $L/2$. As shown in Table 5, RFold performs unexpectedly well on these long-range base pairing predictions. We can also find that UFold performs better in long-range cases than the complete cases. The possible reason may come from the U-Net model architecture that learns multi-scale features. RFold significantly improves UFold in all the metrics by large margins, demonstrating its strong predictive ability.

Table 4: Results on bpRNA-TS0 set.

| Method | Precision | Recall | F1 |
|---|---|---|---|
| Mfold | 0.501 | 0.627 | 0.538 |
| E2Efold | 0.140 | 0.129 | 0.130 |
| RNAstructure | 0.494 | 0.622 | 0.533 |
| RNAsoft | 0.497 | 0.626 | 0.535 |
| RNAfold | 0.494 | 0.631 | 0.536 |
| Contextfold | 0.529 | 0.607 | 0.546 |
| LinearFold | 0.561 | 0.581 | 0.550 |
| MXfold2 | 0.519 | 0.646 | 0.558 |
| Externafold | 0.516 | 0.666 | 0.563 |
| CONTRAfold | 0.528 | 0.655 | 0.567 |
| SPOT-RNA | 0.594 | **0.693** | 0.619 |
| UFold | 0.521 | 0.588 | 0.553 |
| RFold | **0.692** | 0.635 | **0.644** |

Table 5: Results on long-range bpRNA.

| Method | Precision | Recall | F1 |
|---|---|---|---|
| Mfold | 0.315 | 0.450 | 0.356 |
| RNAstructure | 0.299 | 0.428 | 0.339 |
| RNAsoft | 0.310 | 0.448 | 0.353 |
| RNAfold | 0.304 | 0.448 | 0.350 |
| Contextfold | 0.332 | 0.432 | 0.363 |
| LinearFold | 0.281 | 0.355 | 0.305 |
| MXfold2 | 0.318 | 0.450 | 0.360 |
| Externafold | 0.308 | 0.458 | 0.355 |
| CONTRAfold | 0.306 | 0.439 | 0.349 |
| SPOT-RNA | 0.361 | 0.492 | 0.403 |
| UFold | 0.543 | 0.631 | 0.584 |
| RFold | **0.803** | **0.765** | **0.701** |

## 5.4 INFERENCE TIME COMPARISON

We compared the running time of various methods for predicting RNA secondary structures using the RNAStralign testing set with the same experimental setting and the hardware environment as in (Fu et al., 2022). The results are presented in Table 6, which shows the average inference time per sequence. The fastest energy-based method, LinearFold, takes about 0.43s for each sequence. The learning-based baseline, UFold, takes about 0.16s. RFold has the highest inference speed, costing only about 0.02s per sequence. In particular, RFold is about eight times faster than UFold and sixteen times faster than MXfold2. Its fast inference time is due to the simple sequence-to-map transformation.

Table 6: Inference time on the RNAStralign.

| Method | Time |
|---|---|
| CDPfold (Tensorflow) | 300.11 s |
| RNAstructure (C) | 142.02 s |
| CONTRAfold (C++) | 30.58 s |
| Mfold (C) | 7.65 s |
| Eternafold (C++) | 6.42 s |
| RNAsoft (C++) | 4.58 s |
| RNAfold (C) | 0.55 s |
| LinearFold (C++) | 0.43 s |
| SPOT-RNA(Pytorch) | 77.80 s (GPU) |
| E2Efold (Pytorch) | 0.40 s (GPU) |
| MXfold2 (Pytorch) | 0.31 s (GPU) |
| UFold (Pytorch) | 0.16 s (GPU) |
| RFold (Pytorch) | **0.02 s** (GPU) |

## 5.5 ABLATION STUDY

**Decoupled Optimization**  To validate the effectiveness of our proposed decoupled optimization, we conduct an experiment that replaces them with other strategies. The results are summarized in Table 7, where RFold-E and RFold-S denote our model with the strategies of E2Efold and SPOT-RNA, respectively. We ignore the recent UFold because it follows exactly the same strategy as E2Efold. We also report the validity which is a sample-level metric evaluating whether all the constraints are satisfied. Though RFold-E has comparable performance in the first three metrics with ours, many of its predicted structures are invalid. The strategy of SPOT-RNA has incorporated no constraint

that results in its low validity. Moreover, its strategy seems to not fit our model well, which may be caused by the simplicity of our RFold model.

Table 7: Ablation study on optimization strategies (RNAStralign testing set).

| Method | Precision | Recall | F1 | Validity |
|--------|-----------|--------|-------|----------|
| RFold | **0.981** | **0.973** | **0.977** | **100.00**% |
| RFold-E | 0.888 | 0.906 | 0.896 | 50.31% |
| RFold-S | 0.223 | 0.988 | 0.353 | 0.00% |

Table 8: Ablation study on pre-processing strategies (RNAStralign testing set).

| Method | Precision | Recall | F1 | Time |
|--------|-----------|--------|-------|--------|
| RFold | **0.981** | **0.973** | **0.977** | 0.0167 |
| RFold-U | 0.875 | 0.941 | 0.906 | 0.0507 |
| RFold-SS | 0.886 | 0.945 | 0.913 | **0.0158** |

**Seq2map Attention**  We also conduct an experiment to evaluate the proposed Seq2map attention. We replace the Seq2map attention with the hand-crafted features from UFold and the outer concatenation from SPOT-RNA, which are denoted as RFold-U and RFold-SS, respectively. In addition to performance metrics, we also report the average inference time for each RNA sequence to evaluate the model complexity. We summarize the result in Table 8. It can be seen that RFold-U takes much more inference time than our RFold and RFold-SS due to the heavy computational cost when loading and learning from hand-crafted features. Moreover, it is surprising to find that RFold-SS has a little better performance than RFold-U, with the least inference time for its simple outer concatenation operation. However, neither RFold-U nor RFold-SS can provide informative representations.

## 5.6 VISUALIZATION

We visualize two examples predicted by RFold and UFold in Fig. 6. The corresponding F1 scores are denoted at the bottom of each plot. The first secondary structures is a simple example of a nested structure. It can be seen that UFold may fail in such a case. The second secondary structures is much more difficult that contains over 300 bases of the non-nested structure. While UFold fails in such a complex case, RFold can predict the structure accurately. Due to the limited space, we provide more visualization comparisons in Appendix F.

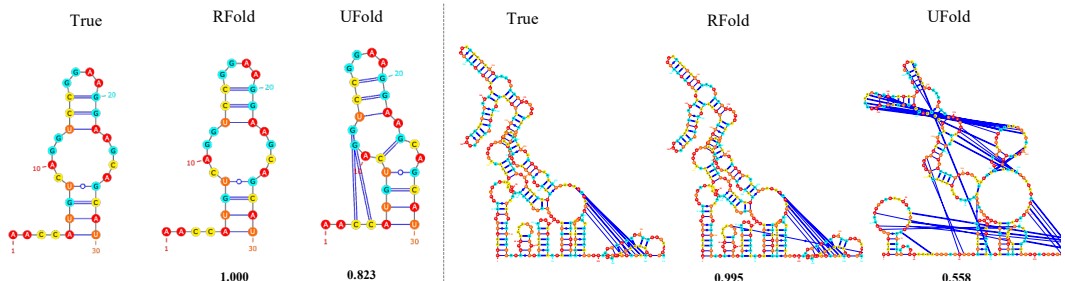

Figure 6: Visualization of the true and predicted structures.

## 6 CONCLUSION AND LIMITATIONS

In this study, we present RFold, a simple yet effective learning-based model for RNA secondary structure prediction. We propose decoupled optimization to replace the complicated post-processing strategies while incorporating constraints for the output. Seq2map attention is proposed for sequence-to-map transformation, which can automatically learn informative representations from a single sequence without extensive pre-processing operations. Comprehensive experiments demonstrate that RFold achieves competitive performance with faster inference speed.

The limitations of RFold primarily revolve around its stringent constraints. This strictness in constraints implies that RFold is cautious in predicting interactions, leading to higher precision but possibly at the cost of missing some true interactions that more lenient models might capture. Though we have provided a naive solution in Appendix D, it needs further studies to obtain a better strategy that leads to more balanced precision-recall trade-offs and more comprehensive structural predictions.

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

## A COMPARISON OF MAINSTREAM RNA SECONDARY STRUCTURE PREDICTION METHODS

We compare our proposed method RFold with several other leading RNA secondary structure prediction methods and summarize the results in Table 9. RFold satisfies all three constraints (a)-(c) for valid RNA secondary structures, while the other methods do not fully meet some of the constraints. RFold utilizes a sequence-to-map attention mechanism to capture long-range dependencies, whereas SPOT-RNA simply concatenates pairwise sequence information and E2Efold/UFold uses hand-crafted features. In terms of prediction accuracy on the RNAStralign benchmark test set, RFold achieves the best F1 score of 0.977, outperforming SPOT-RNA, E2Efold and UFold by a large margin. Regarding the average inference time, RFold is much more efficient and requires only 0.02 seconds to fold the RNAStralign test sequences. In summary, RFold demonstrates superior performance over previous methods for RNA secondary structure prediction in both accuracy and speed.

Table 9: Comparison between RNA secondary structure prediction methods and RFold.

| Method | SPOT-RNA | E2Efold | UFold | RFold |
|---|---|---|---|---|
| pre-processing | pairwise concat | pairwise concat | hand-crafted | seq2map attention |
| backbone model | ResNet + LSTM | Transformer | U-Net | U-Net |
| post-processing | × | unrolled algorithm | unrolled algorithm | decoupled optimization |
| constraint (a) | × | ✓ | ✓ | ✓ |
| constraint (b) | × | ✓ | ✓ | ✓ |
| constraint (c) | × | × | × | ✓ |
| F1 score | 0.711 | 0.821 | 0.915 | **0.977** |
| Inference time | 77.80 s | 0.40 s | 0.16 s | **0.02** s |

## B EXPERIMENTAL DETAILS

**Datasets**   We use three benchmark datasets: (i) RNAStralign (Tan et al., 2017), one of the most comprehensive collections of RNA structures, is composed of 37,149 structures from 8 RNA types; (ii) ArchiveII (Sloma & Mathews, 2016), a widely used benchmark dataset in classical RNA folding methods, containing 3,975 RNA structures from 10 RNA types; (iii) bpRNA (Singh et al., 2019), is a large scale benchmark dataset, containing 102,318 structures from 2,588 RNA types. (iv) bpRNA-new (Sato et al., 2021), derived from Rfam 14.2 (Kalvari et al., 2021), containing sequences from 1500 new RNA families.

**Baselines**   We compare our proposed RFold with baselines including energy-based folding methods such as Mfold (Zuker, 2003), RNAsoft (Andronescu et al., 2003), RNAfold (Lorenz et al., 2011), RNAstructure (Mathews & Turner, 2006), CONTRAfold (Do et al., 2006), Contextfold (Zakov et al., 2011), and LinearFold (Huang et al., 2019); learning-based folding methods such as SPOT-RNA (Singh et al., 2019), Externafold (Wayment-Steele et al., 2021), E2Efold (Chen et al., 2019), MXfold2 (Sato et al., 2021), and UFold (Fu et al., 2022).

**Metrics**   We evaluate the performance by precision, recall, and F1 score, which are defined as:

$$\text{Precision} = \frac{\text{TP}}{\text{TP} + \text{FP}}, \ \text{Recall} = \frac{\text{TP}}{\text{TP} + \text{FN}}, \ \text{F1} = 2\frac{\text{Precision} \cdot \text{Recall}}{\text{Precision} + \text{Recall}}, \tag{19}$$

where $\text{TP}, \text{FP},$ and $\text{FN}$ denote true positive, false positive and false negative, respectively.

**Implementation details**   Following the same experimental setting as (Fu et al., 2022), we train the model for 100 epochs with the Adam optimizer. The learning rate is 0.001, and the batch size is 1 for sequences with different lengths.

## C   RESULTS ON RNA STRUCTURES WITH PSEUDOKNOTS

Following E2Efold and UFold, we count the number of pseudoknotted sequences that are predicted as pseudoknotted and report this count as true positive. We pick all sequences containing pseudoknots from the RNAStralign test dataset. The results are as follows:

Table 10: Results on RNA structures with pseudoknots.

| Method | Precision | Recall | F1 Score |
|---|---|---|---|
| RNAstructure | 0.778 | 0.761 | 0.769 |
| SPOT-RNA | 0.677 | 0.978 | 0.800 |
| E2Efold | 0.844 | 0.990 | 0.911 |
| UFold | 0.962 | 0.990 | 0.976 |
| RFold | 0.971 | 0.993 | 0.982 |

As the result demonstrates, RFold consistently surpasses UFold across all three metrics, indicating the effectiveness of our proposed approach.

## D   DISCUSSION ON ABNORMAL SAMPLES

Although we have illustrated three hard constraints in 3.2, there exist some abnormal samples that do not satisfy these constraints in practice. We have analyzed the datasets used in this paper and found that there are some abnormal samples in the testing set that do not meet these constraints. The ratio of valid samples in each dataset is summarized in the table below:

Table 11: The ratio of valid samples in the datasets.

| Dataset | RNAStralign | ArchiveII | bpRNA |
|---|---|---|---|
| Validity | 93.05% | 96.03% | 96.51% |

As shown in Table 7, RFold forces the validity to be 100.00%, while other methods like E2Efold only achieve about 50.31%. RFold is more accurate than other methods in reflecting the real situation.

Nevertheless, we provide a soft version of RFold to relax the strict constraints. A possible solution to relax the rigid procedure is to add a checking mechanism before the Rol-Col Argmax function in the inference. Specifically, if the confidence given by the Rol-Col Softmax is low, we do not perform Rol-Col Argmax and assign more base pairs. It can be implemented as the following pseudo-code:

```
y_pred = row_col_softmax(y)
int_one = row_col_argmax(y_pred)

# get the confidence for each position
conf = y_pred * int_one
all_pos = conf > 0.0

# select reliable position
conf_pos = conf > thr1

# select unreliable position with the full row and column
uncf_pos = get_unreliable_pos(all_pos, conf_pos)

# assign "1" for the positions with the confidence higher than thr2
# note that thr2 < thr1
y_pred[uncf_pos] = (y_pred[uncf_pos] > thr2).float()
int_one[uncf_pos] = y_pred[uncf_pos]
```

We conduct experiments to compare the soft-RFold and the original version of RFold in the RNAS-tralign dataset. The results are summarized in the Table 12. It can be seen that soft-RFold improves

Table 12: The results of soft-RFold and RFold on the RNAStralign.

| Method | Precision | Recall | F1 |
|--------|-----------|--------|-----|
| RFold | 0.981 | 0.973 | 0.977 |
| soft-RFold | 0.978 | 0.974 | 0.976 |

Table 13: The results of soft-RFold and RFold on the abnormal samples on the RNAStralign.

| Method | Precision | Recall | F1 |
|--------|-----------|--------|-----|
| RFold | 0.956 | 0.860 | 0.905 |
| soft-RFold | 0.949 | 0.889 | 0.918 |

the recall metric by a small margin. The minor improvement may be because the number of abnormal samples is small.

We then select those samples that do not obey the three constraints to further analyse the performance. The total number of such samples is 179. It can be seen that soft-RFold can deal with abnormal samples well. The improvement of the recall metric is more obvious.

# E  PROOFS OF THEOREMS

## E.1  PROOF OF THEOREM 1

**Theorem 1.** Given a symmetric matrix $\widehat{\boldsymbol{H}} \in \mathbb{R}^{L \times L}$, the matrix Row-Col-Argmax($\widehat{\boldsymbol{H}}$) is also a symmetric matrix.

*Proof:* From Eq. 10 and Eq. 11, we can know that:

$$
\begin{aligned}
&\text{Row-Col-Argmax}(\widehat{\boldsymbol{H}}_{ij}) = 1, \\
&\text{if } \max\{\{\widehat{\boldsymbol{H}}_{ik}\}_{k=1}^{L} \cup \{\widehat{\boldsymbol{H}}_{kj}\}_{k=1}^{L}\} = \widehat{\boldsymbol{H}}_{ij},
\end{aligned}
\tag{20}
$$

Then, we can infer that:

$$
\begin{aligned}
&\text{Row-Col-Argmax}(\widehat{\boldsymbol{H}}_{ji}) = 1, \\
&\text{if } \max\{\{\widehat{\boldsymbol{H}}_{jk}\}_{k=1}^{L} \cup \{\widehat{\boldsymbol{H}}_{ki}\}_{k=1}^{L}\} = \widehat{\boldsymbol{H}}_{ji},
\end{aligned}
\tag{21}
$$

As $\widehat{\boldsymbol{H}}$ is a symmetric matrix, $\widehat{\boldsymbol{H}}_{jk} = \widehat{\boldsymbol{H}}_{kj}$ and $\widehat{\boldsymbol{H}}_{ki} = \widehat{\boldsymbol{H}}_{ik}$. Thus, Row-Col-Argmax($\widehat{\boldsymbol{H}}_{ji}$) can be rewritten as:

$$
\begin{aligned}
&\text{Row-Col-Argmax}(\widehat{\boldsymbol{H}}_{ji}) = 1, \\
&\text{if } \max\{\{\widehat{\boldsymbol{H}}_{kj}\}_{k=1}^{L} \cup \{\widehat{\boldsymbol{H}}_{ik}\}_{k=1}^{L}\} = \widehat{\boldsymbol{H}}_{ij},
\end{aligned}
\tag{22}
$$

It can be seen that only if $\max\{\{\widehat{\boldsymbol{H}}_{kj}\}_{k=1}^{L} \cup \{\widehat{\boldsymbol{H}}_{ik}\}_{k=1}^{L}\} = \widehat{\boldsymbol{H}}_{ij} = \widehat{\boldsymbol{H}}_{ji}$, then $\widehat{\boldsymbol{H}}_{ij} = \widehat{\boldsymbol{H}}_{ji} = 1$.

Thus, Row-Col-Argmax($\widehat{\boldsymbol{H}}$) is also a symmetric matrix.

## E.2  PROOF OF THEOREM 2

**Theorem 2.** Given a symmetric matrix $\widehat{\boldsymbol{H}} \in \mathbb{R}^{L \times L}$, the matrix Row-Col-Softmax($\widehat{\boldsymbol{H}}$) is also a symmetric matrix.

*Proof:* $\forall i, j \in \{1, ..., L\}$,

$$
\begin{aligned}
&\text{Row-Col-Softmax}(\widehat{\boldsymbol{H}}_{ji}) \\
&= \frac{1}{2}\left(\frac{\exp(\widehat{\boldsymbol{H}}_{ji})}{\sum_{k=1}^{L}\exp(\widehat{\boldsymbol{H}}_{jk})} + \frac{\exp(\widehat{\boldsymbol{H}}_{ji})}{\sum_{k=1}^{L}\exp(\widehat{\boldsymbol{H}}_{ki})}\right) \\
&= \frac{1}{2}\left(\frac{\exp(\widehat{\boldsymbol{H}}_{ij})}{\sum_{k=1}^{L}\exp(\widehat{\boldsymbol{H}}_{kj})} + \frac{\exp(\widehat{\boldsymbol{H}}_{ij})}{\sum_{k=1}^{L}\exp(\widehat{\boldsymbol{H}}_{ik})}\right) \\
&= \text{Row-Col-Softmax}(\widehat{\boldsymbol{H}}_{ij}).
\end{aligned}
\tag{23}
$$

# F  VISUALIZATION

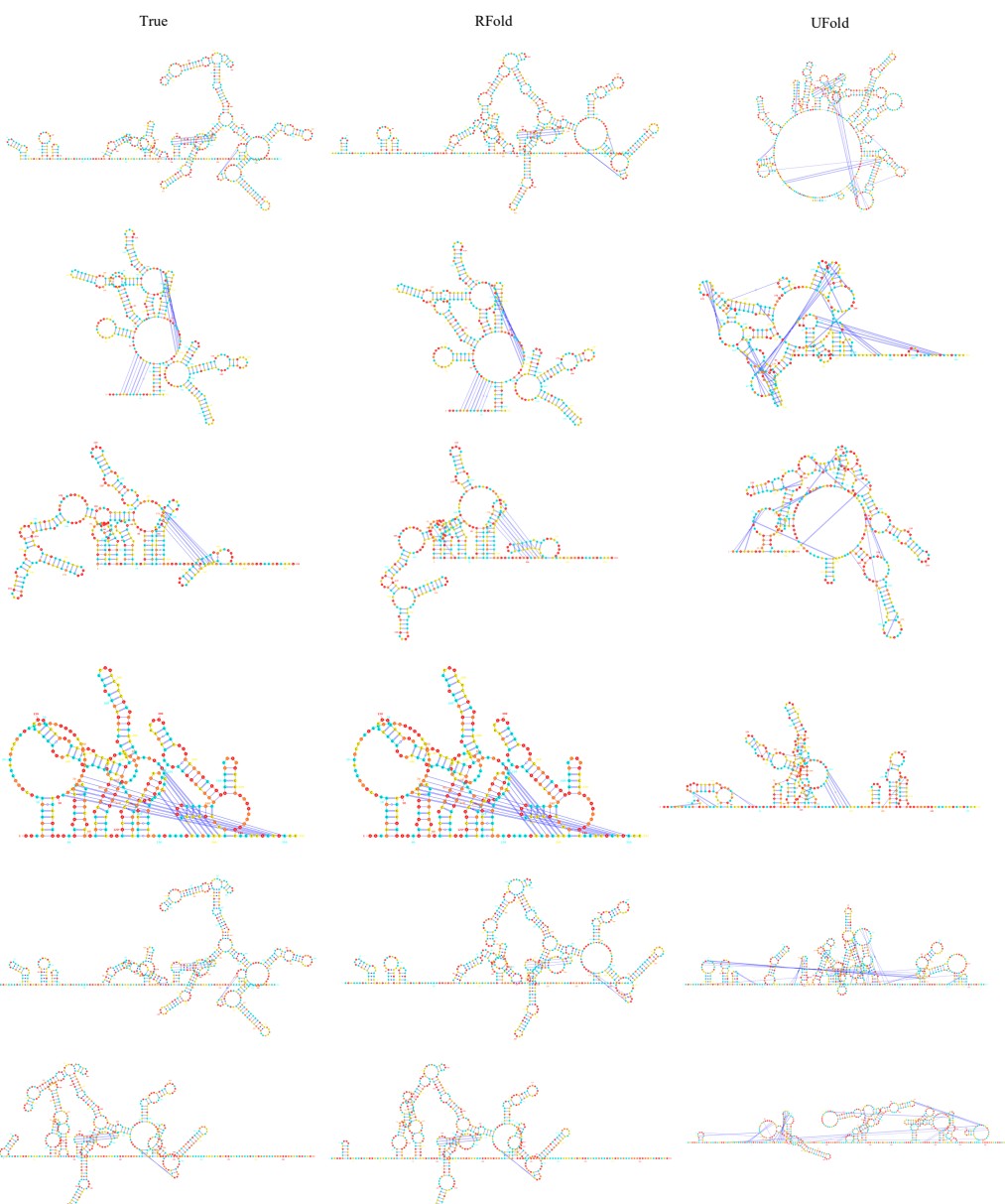

Figure 7: Visualization of the true and predicted structures.

# G  LOSS CURVE

We present the training and validation loss curves of RFold on the RNAStralign dataset in Fig. 8. Note that in our log files, the loss values are recorded in four decimal places. The training loss begins at a higher value and exhibits a consistent decrease across epochs, indicating effective learning. Similarly, the validation loss also shows a decreasing trend throughout the epochs. Although there are minor fluctuations, the overall trajectory is toward lower loss values, suggesting that the model is generalizing effectively on the validation data.

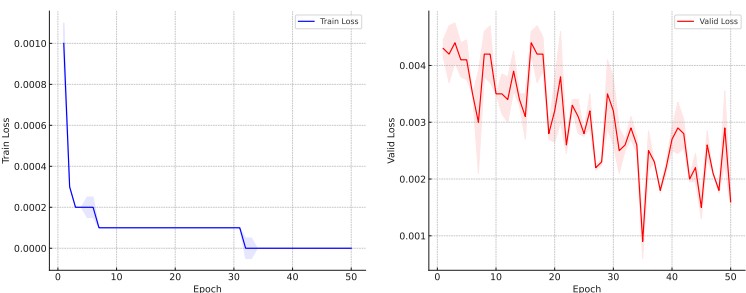

Figure 8: The training and validation loss curve on RNAStralign dataset.

