# OpenReview forum: "RFold: RNA Secondary Structure Prediction with Decoupled Optimization"
_ICLR.cc/2024/Conference — Submitted to ICLR 2024_

### Official Review · Reviewer_G4jZ · 2023-10-24

**Soundness:** 2 fair
**Presentation:** 1 poor
**Contribution:** 2 fair
**Rating:** 3
**Confidence:** 5

**Summary:**

In this paper, the authors propose RFold, a simple yet effective RNA secondary structure prediction in an end-to-end manner. Speciﬁcally, a decoupled optimization process that decomposes the vanilla constraint satisfaction problem into row-wise and column-wise optimization is introduced to simplify the solving process while guaranteeing the validity of the output. Besides, RFlod adopts attention maps as informative representations to automatically learn the pair-wise interactions of the nucleotide bases instead of using hand-crafted features to perform data pre-processing.

**Strengths:**

1. The article closely links RNA-related issues with the ICLR community.

**Weaknesses:**

1. Presentation:
    - Grammar issues, such as in the sentence: "The general deep-learning-based RNA secondary structure prediction methods into three key parts," which is unclear.
    - Incorrect citation format; the ICLR official template recommends the use of \citep, but it seems the author used \cite, causing the citation information to mix with the text, which hampers readability.
    - Inconsistent abbreviation of equations; some use "Eq." while others use "Equ." (e.g., in the last line of page 5).

2. The experimental results are not sufficiently detailed. The article emphasizes the validity of the proposed method's prediction results but lacks validity results for the baselines.

**Questions:**

1. I gave this article a rejection recommendation, largely due to its poor presentation. In addition to the specific issues mentioned in the Weaknesses section, the explanation of the method is not very clear. For example, I'm not very clear about how Eq. 9 is derived. From my understanding, $\mathbf{M}$ should represent the ground-truth, and $S_r, S_c$ constitute the decomposition of M. Then, is there any necessity to optimize $S_r$ and $S_c$?
2. In Equation 13, there is no rigorous explanation provided for why the Row-Col Softmax function is used to approximate the Row-Col Argmax function for training.

If the author can effectively address my concerns, especially those related to presentation, I will reconsider this article and possibly revise my evaluation.

---

> ### Author Response · Authors · 2023-11-13
>
> Dear Reviewer G4jZ,
>
> Thank you for your professional comments. We apologize for the issues in the presentation. We have addressed these issues in the revised manuscript, highlighting the changes in blue font.
>
> **Q1** The validity results.
>
> **A1** Thank you for your kind suggestion. In the ablation study, we compared different post-processing strategies, as shown in Table 7. To eliminate the influence of the backbone model and preprocessing methods, we used RFold as an example. We compared RFold-E, which employs an unrolled algorithm, with RFold-S, which does not use any post-processing. We found that the validity of RFold's post-processing strategies is significantly better than that of others.
>
> **Q2** The explanation of the method.
>
> **A2** Yes, $\boldsymbol{M}$ represents the ground-truth, and $S_r, S_c$ constitute the decomposition of $\boldsymbol{M}$. However, directly optimizing for the final solution is infeasible. Our proposed decoupled optimization approach utilizes the symmetry of the matrix $\widehat{H}$ to decouple the optimization into row-wise and column-wise components. Optimizing $S_r, S_c$​ separately allows for a more manageable and efficient computation process.
>
> **Q3** Why the Row-Col Softmax function is used to approximate the Row-Col Argmax function for training?
>
> **A3** Thank you for your insightful comment. The use of the Row-Col Softmax function to approximate the Row-Col Argmax function in the RFold model for RNA secondary structure prediction is primarily due to the differentiability requirements of the training process.
>
> The Argmax function, which is used in the inference phase of RFold, is non-differentiable. On the other hand, the Softmax function is a differentiable approximation of the Argmax function. It provides a "soft" version of the maximum, assigning probabilities to each element in such a way that larger values get higher probabilities。The Row-Col Softmax function serves as a differentiable proxy during training. It allows the model to learn the appropriate parameters by providing gradient information which is not possible with the Row-Col Argmax function.
>
> Once the model is trained, during the inference phase, the Row-Col Softmax function can be replaced with the Row-Col Argmax function. This is because inference does not involve backpropagation or gradient computations, and the Argmax function provides a clear, decisive prediction that is necessary for the final RNA structure prediction.

---

> > ### Author Response · Authors · 2023-11-20
> >
> > Dear Reviewer,
> >
> > We sincerely appreciate your valuable suggestions which help us improve our manuscript. We have responded to your questions point by point.
> >
> > Unlike previous years, there will be no second stage of author-reviewer discussions this year. As the deadline for discussion is approaching, please do not hesitate to reach out if you have any further inquiries.
> >
> > We kindly request your attention and ask if there are additional concerns we can address for you to consider raising the score.
> >
> > Thank you for your time and consideration.
> >
> > Sincerely,
> >
> > Authors

---

> ### Author Response · Authors · 2023-11-23
> **Looking for your response**
>
> Dear reviewer G4jZ,
>
> We would like to kindly remind you that ICLR allows for revisions, and the textual issues you mentioned have been considered minors in other reviews, as we are also reviewers of mutiple papers. In the initial round of review, you graciously expressed that:
>
> "If the author can effectively address my concerns, especially those related to presentation, I will reconsider this article and possibly revise my evaluation."
>
> We have diligently revised the manuscript based on your suggestions. As the rebuttal deadline is approaching, we would greatly appreciate hearing your response. Thanks for your valuable review and comments again. We have addressed all your concerns and questions. If you have any other questions, please feel free to discuss with us.
>
> Best,
>
> Authors

---

### Official Review · Reviewer_BXWa · 2023-11-01

**Soundness:** 3 good
**Presentation:** 3 good
**Contribution:** 2 fair
**Rating:** 5
**Confidence:** 3

**Summary:**

The paper proposes RFold, a method for RNA secondary structure prediction. RFold introduces decoupled optimization that decomposes the constraint satisfaction problem into separate row-wise and column-wise optimizations. This simplifies optimization while guaranteeing valid outputs. RFold also employs Seq2map attention to automatically learn sequence representations, avoiding manual feature engineering. Experiments show RFold matches state-of-the-art approaches on benchmarks while having faster inference. Ablations validate the optimization and attention contributions.

**Strengths:**

The strength of RFold is its simple yet effective approach for RNA structure prediction. The post-processing step simplifies satisfying complex structural constraints. The pre-processing step automatically learns useful sequence representations without feature engineering. Together these enable high performance prediction with fast inference. Rigorous experiments demonstrate RFold matches or exceeds state-of-the-art methods across multiple benchmarks. The ablation studies clearly validate the benefits of the proposed techniques.

**Weaknesses:**

A key weakness is the technical approaches seem incremental, lacking major deep learning innovations. The decoupled optimization and Seq2map attention offer straightforward extensions to UFold. This suggests the work may be better suited for a venue focused on the specific domain.

**Questions:**

How is Equation 10 obtained with the proposed decoupled optimization?

---

> ### Author Response · Authors · 2023-11-13
>
> Dear Reviewer BXWa,
>
> Thank you for your constructive comments!
>
> **Q1** A key weakness is the technical approaches seem incremental, lacking major deep learning innovations. The decoupled optimization and Seq2map attention offer straightforward extensions to UFold. This suggests the work may be better suited for a venue focused on the specific domain.
>
> **A1** **The seminal work, E2Efold [1], was accepted by ICLR 2020 as one of the 48 oral presentations.** As shown in the methodology comparison table below, compared to SPOT-RNA, E2Efold modified the backbone model to a Transformer and introduced an unrolling algorithm for post-processing. Although promising, E2Efold did not perfectly address the problem of three constraints.
>
> By employing novel decoupled optimization, **our proposed RFold is the first work to fully satisfy all constraints**. Furthermore, while UFold attempted to enhance performance with more complex pre-processing, our approach utilizes a straightforward seq2map attention to automatically capture pairwise features, thereby improving inference speed. We respect previous works, but believe RFold offers unique contributions.
>
> | Method    | SPOT-RNA      | E2Efold               | UFold                 | RFold                 |
> |----------------------------------|---------------|-----------------------|-----------------------|-----------------------|
> | pre-processing                   | pairwise concat | pairwise concat       | hand-crafted          | seq2map attention     |
> | backbone model                   | ResNet + LSTM | Transformer           | U-Net                 | U-Net                 |
> | post-processing                  | ×             | unrolled algorithm    | unrolled algorithm    | decoupled optimization|
> | constraint (a)                   | ×             | ✔                     | ✔                     | ✔                     |
> | constraint (b)                   | ×             | ✔                     | ✔                     | ✔                     |
> | constraint (c)                   | ×             | ×                     | ×                     | ✔                     |
> | RNAStralign F1 score | 0.711         | 0.821                 | 0.915                 | **0.977**             |
> | Inference time                   | 77.80 s       | 0.40 s                | 0.16 s                | **0.02** s            |
>
>
> **Q2** How is Equation 10 obtained with the proposed decoupled optimization?
>
> **A2** Our proposed decoupled optimization approach in RFold simplifies the complex task of RNA secondary structure prediction by dividing it into separate row-wise and column-wise optimizations. This method effectively manages the complexity by handling these aspects independently and then integrating them to form a comprehensive solution. The process relies on the symmetry of the matrix $\widehat{H}$. We utilize Argmax functions for each sub-problem due to their suitability in selecting maximal values while maintaining structure constraints, though they are not the sole method available for such optimizations.
>
> [1] RNA Secondary Structure Prediction By Learning Unrolled Algorithms, ICLR 2020.

---

> > ### Author Response · Authors · 2023-11-20
> >
> > Dear Reviewer,
> >
> > Thank you for the time and effort you have invested in reviewing our work. We have responded to your questions point by point.
> >
> > As the deadline for discussion is approaching, we kindly request that you inform us if there is a possibility of raising the score.
> >
> > Best regards,
> >
> > Authors.

---

> > > ### Comment · Reviewer_BXWa · 2023-11-23
> > >
> > > Thanks for the response to my concerns and questions. I acknowledge that I have read the authors' comments. I have increased my score to 5 correspondingly.

---

> > > > ### Author Response · Authors · 2023-11-23
> > > >
> > > > We are immensely delighted to hear that concerns were addressed and the score has been improved to 5. We would respectfully inquire if there might be any additional opportunities for us to further improve our manuscript and potentially increase the rating.
> > > >
> > > > Once again, we express our sincere appreciation for your valuable feedback. We would eagerly welcome any further guidance you may provide. Thank you for your patience, review efforts, and good questions!

---

### Official Review · Reviewer_Y2Rp · 2023-11-01

**Soundness:** 3 good
**Presentation:** 4 excellent
**Contribution:** 3 good
**Rating:** 6
**Confidence:** 3

**Summary:**

This paper proposes a decoupled optimization approach for the RNA folding problem. Through decoupled optimization formulation of the mapping matrix learning, the RNA folding prediction becomes much faster. Comprehensive experiments demonstrate the superiority of the proposed method compared to bunch of baseline methods.

**Strengths:**

(1) The writing is well-organized and the core idea is clearly presented. In general, although this reviewer has not read too much RNA paper before, the problem formulation and key algorithm design can be easily captured;

(2) This paper conducts very comprehensive experiments to demonstrate the effectiveness of the proposed approach. This reviewer firmly believes the proposed method can bring some improvements to this domain of research.

**Weaknesses:**

It seems that the major point of the proposed method is mainly about the optimization formulation of the assignment matrix learning while the seq2map network architecture and the mapping matrix formulation are mainly inherited from previous works. So probably the novelty of the proposed method is a little bit limited. However, this reviewer is not very familiar with RNA folding frontier research, thus it is not very fair for this reviewer to evaluate its novelty and significance.

**Questions:**

Does the Rol-Column-Softmax operation have a training instability issue? Since this operation is very similar to the Sinkhorn operation and the Sinkhorn algorithm has training stability issues. This reviewer afraid that the proposed method also has the same drawback.

---

> ### Author Response · Authors · 2023-11-13
>
> Dear Reviewer Y2Rp,
>
> Thank you for your thoughtful and inspiring comment!
>
> **Q1** The proposed method is mainly about the optimization formulation of the assignment matrix learning while the seq2map network architecture and the mapping matrix formulation are mainly inherited from previous works.
>
> **A1** Thank you for your meticulous review! We would like to claim that both the optimization formulation and the seq2map attention are our main contributions. As stated in the introduction, we decompose deep learning-based RNA secondary structure prediction methods into three key components: preprocessing, the backbone model, and post-processing.
> * The pre-processing step means projecting the 1D sequence into 2D matrix. (**1D -> discrete 2D**)
> * The backbone model learns from the 2D matrix and then outputs a hidden matrix of continuous values. (**discrete 2D -> continuous 2D**)
> * The post-processing step converts the hidden matrix into a contact map, which is a matrix of discrete 0/1 values. (**continuous 2D -> discrete 2D**)
>
> The methodology comparison is shown in the table below:
>
> | Method                           | SPOT-RNA      | E2Efold               | UFold                 | RFold                 |
> |----------------------------------|---------------|-----------------------|-----------------------|-----------------------|
> | pre-processing                   | pairwise concat | pairwise concat       | hand-crafted          | seq2map attention     |
> | backbone model                   | ResNet + LSTM | Transformer           | U-Net                 | U-Net                 |
> | post-processing                  | ×             | unrolled algorithm    | unrolled algorithm    | decoupled optimization|
> | constraint (a)                   | ×             | ✔                     | ✔                     | ✔                     |
> | constraint (b)                   | ×             | ✔                     | ✔                     | ✔                     |
> | constraint (c)                   | ×             | ×                     | ×                     | ✔                     |
> | RNAStralign F1 score | 0.711         | 0.821                 | 0.915                 | **0.977**             |
> | Inference time                   | 77.80 s       | 0.40 s                | 0.16 s                | **0.02** s            |
>
> It can be observed that the only similarity between RFold and previous methods is the use of U-Net as the backbone model, similar to UFold. Furthermore, the most significant difference of RFold compared to earlier methods is that its predictions for secondary structures are guaranteed to satisfy all three constraints.
>
>
> **Q2** Does the Rol-Column-Softmax operation have a training instability issue?
>
> **A2** Thank you for your insightful question! We have added the loss curve plot in **Appendix G of the revised manuscript**. It can be observed that the training loss decreased steadily across the training epochs. The possible reasons for this may include: (i) The Softmax function, used in the Row-Column-Softmax, is generally numerically stable; (ii) The objective of the Row-Column-Softmax is not to make the matrix doubly stochastic (as in the Sinkhorn algorithm) but to approximate the Row-Col-Argmax operation in a differentiable manner for training purposes.

---

> > ### Author Response · Authors · 2023-11-20
> >
> > Dear Reviewer,
> >
> > Thank you for your insightful and helpful comments once again. We greatly appreciate your feedback. We have carefully responded to each of your questions point-by-point.
> >
> > Unlike previous years, there will be no second stage of author-reviewer discussions this year. As the deadline for the discussion is approaching, we kindly request you to inform us if you have any additional questions.
> >
> > Best regards,
> >
> > Authors.

---

### Official Review · Reviewer_fGRe · 2023-11-04

**Soundness:** 3 good
**Presentation:** 4 excellent
**Contribution:** 4 excellent
**Rating:** 6
**Confidence:** 2

**Summary:**

This paper proposes an end-to-end deep learning pipeline for RNA secondary structure prediction from the input 1D sequence. Compared to previous deep learning pipelines, the proposed one simplifies the manually-crafted data pre-processing step with the learned representation and the result post-processing step through reformulating and approximating the loss function. On standard benchmarks, the proposed method significantly outperforms current SOTA methods in both accuracy and speed. The ablation studies clearly demonstrate the effectiveness of the proposed design choices.

**Strengths:**

- The paper is well-written. The "preliminary" section clearly summarizing existing works and the figures are helpful for understanding.

- The proposed method has a simple yet effective design.

- The proposed method significantly outperforms the previous SOTA on the standard benchmarks.

- The ablation studies are helpful to verify the design choices for the data pre-processing and result post-processing.

**Weaknesses:**

On the high-level, the proposed method employs a simple trick to decode M from H: compute the row-argmax and col-argmax, and then Hadamard product them, where the output is guaranteed to be a symmetric 0-1 matrix satisfying the constraints. Overall, the paper does a good job to justify this simple approach by reformulating the optimization objective and approximating it to the train deep learning model. However, there are some details that may need modification.

- The solution to Eq (6) is not Eq (10). Here, H is the unconstrained output and \hat H can have negative values. (1) If \hat H is all non-positive (0 on the diagonal), the optimal solution to Eq (6) is M=0. (2) Even if \hat H is non-negative, consider \hat H = [5,4;4,1]. The row-col-argmax(\hat H) = [1,0;0,0] while the optimal for Eq (6) is [1,0;0,1].

- Suppose Eq (10) is correct, it is unclear how good is the approximation in Eq (14). (1) It'll be great to compare with the approximation (row-softmax \odot col-softmax). (2) argmax can been seen as softmax with temperature T->0. Current approximation uses T=1. It'll be great to add an ablation study to examine how good is the softmax approximation.

--------------
Minor
- Eq (1): the order of composition should be swapped. F(x) = G[H(x)]. => F = G\circ H

**Questions:**

- Under Eq (9), what does \otimes refer to? Are S_r and S_c sets of vectors? Maybe better to define S_r and S_c as matrices and use Hadamard product.
- I really like the paper, but the soundness of the formulation and approximation slightly make me concern as pointed out in the weakness section. I'd like to hear authors' explanation on them. One simple change is to tune down the claim "optimal solution" to "a greedy algorithm solution".

---

> ### Author Response · Authors · 2023-11-13
>
> Dear Reviewer fGRe,
>
> We sincerely appreciate your careful and insightful comments!
>
> **Q1** $\boldsymbol{\widehat{H}}$ can have negative values?
>
> **A1** Thank you again for your careful review! We have defined $\boldsymbol{\widehat{H}} = (\boldsymbol{H} \odot \boldsymbol{H}^T) \odot \boldsymbol{\bar{M}}$ below Eq.6. The element-wise multiplication $\boldsymbol{H} \odot \boldsymbol{H}^T$ results in a matrix where each element is the square of the corresponding element in $\boldsymbol{H}$. Since squaring any real number results in a non-negative value, this operation yields a matrix with non-negative elements. $\boldsymbol{\bar{M}} \in \{0,1\}$ contains only non-negative values, typical for a constraint matrix, often containing 0s and 1s to indicate the presence or absence of constraints. Thus, the final element-wise multiplication in $\boldsymbol{\widehat{H}}$ does not introduce any negative values.
>
> **Q2** The row-col-argmax($\boldsymbol{\widehat{H}}$) may be not optimal.
>
> **A2** The example is interesting. However, considering $\boldsymbol{\widehat{H}} = (\boldsymbol{H} \odot \boldsymbol{H}^T) \odot \boldsymbol{\bar{M}}$ and $\boldsymbol{\bar{M}}$ is defined as $\boldsymbol{\bar{M}}_{ij}:=1$ if $x_i x_j \in \mathcal{B}$ and $|i-j| \geq 4$. This example is not suitable for our case because it excludes the diagonal.
>
> **Q3** How good is the approximation in Eq (14)?
>
> **A3** Your understanding that row-col softmax is an approximation of row-col argmax is definitely correct. We should use row-column argmax because the predicted contact matrix is meant to be discrete. However, row-column argmax is not differentiable, which is why we use row-column softmax in the training phase for differentiable optimization.
>
> **Q4** Eq (1): the order of composition should be swapped. F(x) = G[H(x)]. => F = G\circ H.
>
> **A4** Thanks for your careful review. We have fixed this issue in the revised manuscript.
>
> **Q5** Under Eq (9), what does \otimes refer to? Are $S_r$ and $S_c$ sets of vectors? Maybe better to define $S_r$ and $S_c$ as matrices and use Hadamard product.
>
> **A5** We are sorry for the confusion. In our context, $S_r$ and $S_c$ represent vectors, and the symbol $\otimes$ denotes the Hadamard product. The formulation has been refined in the revised manuscript for greater clarity.
>
> **Q6** Tune down the claim "optimal solution" to "a greedy algorithm solution".
>
> **A6** Thank you for your insightful suggestion. We agree that 'a greedy algorithm solution' is suitable because it allows us to decompose the problem into row-wise and column-wise sub-problems. We have refined the description in the revised manuscript.

---

> > ### Author Response · Authors · 2023-11-20
> >
> > Dear Reviewer,
> >
> > Thank you for your insightful and helpful comments once again. We greatly appreciate your feedback. We have carefully responded to each of your questions point-by-point.
> >
> > Unlike previous years, there will be no second stage of author-reviewer discussions this year. As the deadline for the discussion is approaching, we kindly request you to inform us if you have any additional questions.
> >
> > Best regards,
> >
> > Authors.

---

### Author Response · Authors · 2023-11-20

Dear Reviewers,

We sincerely thank you for dedicating time to review our manuscript and for your valuable suggestions.

This year, unlike previous years, there will be no second stage of author-reviewer discussions, and a decision is required by November 22, 2023. Considering that the author-reviewer discussion phase is nearing the end, we hope to confirm whether our responses have sufficiently addressed your concerns.

We have provided detailed replies to your concerns a few days ago, and we hope we have satisfactorily addressed your concerns. If you still need any clarification or have any other concerns, please feel free to contact us and we are happy to continue communicating with you.

Best regards,

Authors

---

### Comment · Area_Chair_tE7y · 2023-11-22
**Discussion Period Ending - Please Engage**

Dear Reviewers,

The authors have responded to the reviews. Please read over the responses and take this opportunity to clarify any further issues before the end of the discussion period today (Nov 22 AOE).

Thanks,
AC

---

### Meta-Review · Area_Chair_tE7y · 2023-12-07

**Metareview:**

This paper introduces a deep learning algorithm for RNA secondary structure prediction that also satisfies base-pairing constraints, unlike previous deep learning based approaches. The key innovations over existing approaches are the constrained optimization approach and seq2map attention mechanism. Experiments show the proposed RFold method outperforms state-of-the-art while also having valid structures; ablation studies show the effectiveness of the two key innovations.

Reviewers appreciated the strong evaluations and clear presentation, though there were concerns about technical novelty and the formulation. The authors provided comprehensive responses that addressed some of these issues, resulting in one reviewer raising their score. During the discussion there were lingering concerns about the technical novelty of the work and reviewers leaned towards rejection.

This paper indeed solves an important issue with previous deep learning methods in terms of base-pairing constraints, but does so by applying known techniques to this specific problem. Hence, in the context of deep learning methodology, the AC tends to agree with the reviewers that there is limited novelty and recommends rejection. That being said, this is nonetheless an important contribution to the RNA secondary structure prediction field and perhaps it is more suited to be published at a domain specific venue like the most recent baseline UFold, where it will reach the right audience.

**Justification For Why Not Higher Score:**

- Limited technical novelty in terms of deep learning makes this more suitable for an domain-specific venue

**Justification For Why Not Lower Score:**

N/A

---

### Decision · Program_Chairs · 2024-01-16

Reject